# MaxMin-Novelty: Maximizing Novelty via Minimizing the State-Action Values in Deep Reinforcement Learning

## Abstract

Reinforcement learning research has achieved high acceleration in its progress starting from the initial installation of deep neural networks as function approximators to learn policies that make sequential decisions in high-dimensional state representation MDPs. While several consecutive barriers have been broken in deep reinforcement learning research (i.e. learning from high-dimensional states, learning purely via self-play), several others still stand. On this line, the question of how to explore in high-dimensional complex MDPs is a well-understudied and ongoing open problem. To address this, in our paper we propose a unique exploration technique based on maximization of novelty via minimization of the state-action value function (MaxMin Novelty). Our method is theoretically well motivated, and comes with zero computational cost while leading to significant sample efficiency gains in deep reinforcement learning training. We conduct extensive experiments in the Arcade Learning Environment with high-dimensional state representation MDPs. We show that our technique improves the human normalized median scores of Arcade Learning Environment by $248\%$ in the low-data regime.

## 1 Introduction

Utilization of deep neural networks as function approximators enabled learning functioning policies in high-dimensional state representation MDPs (Mnih et al., 2015). Following this initial work, the current line of work trains deep reinforcement learning policies to solve highly complex problems from game solving (Hasselt et al., 2016; Schrittwieser et al., 2020) to self driving vehicles (Lan et al., 2020). Yet there are still remaining unsolved problems restricting the current capabilities of deep neural policies.

One of the main intrinsic open problems in deep reinforcement learning research is exploration in high-dimensional state representation MDPs. While prior work extensively studied the exploration problem in bandits and tabular reinforcement learning, and proposed various algorithms and techniques optimal to the tabular form or the bandit setting (Kearns & Singh, 2002; Brafman & Tennenholtz, 2002; Lu & Roy, 2019; Wang et al., 2020; Karnin et al., 2013; Wagenmaker et al., 2022), exploration in deep reinforcement learning remains an open challenging problem.

Despite the provable optimality of these exploration techniques in the tabular or bandit setting, they generally rely strongly on the assumptions of tabular reinforcement learning, and in particular on the ability to record tables of statistical estimates for every state-action pair. Thus, in high-dimensional complex MDPs, for which deep neural networks are used as function approximators, the efficiency and the optimality of exploration methods proposed for tabular settings do not transfer well to deep reinforcement learning exploration. This is primarily due to the increase in the MDP dimensions and the incline in the complexity. Hence, in deep reinforcement learning research still, naive and simple exploration techniques (e.g. $\epsilon$-greedy) are preferred over the optimal tabular techniques (Mnih et al., 2015; Hasselt et al., 2016; Wang et al., 2016; Anschel et al., 2017; Bellemare et al., 2017; Lan et al., 2020).

Sample efficiency in deep neural policies is still one of the main challenging problems restricting research progress in reinforcement learning. The magnitude of the number of samples required to

learn and adapt continuously is one of the main limiting factors preventing current state-of-the-art deep reinforcement learning algorithms from being deployed in many diverse settings, but most importantly one of the main challenges that needs to be dealt with on the way to building general artificial intelligence. In our paper we aim to seek answers for the following questions:

- *Can we explore a high-dimensional state representation MDP more efficiently with zero additional computational cost?*
- *Is there a natural theoretical motivation that can be used to design a zero-cost exploration strategy while achieving high sample efficiency?*

To be able to answer these questions, in our paper we focus on exploration in deep reinforcement learning and make the following contributions:

- We propose a novel exploration technique based on minimizing the state-action value function to increase the information gain from each particular experience acquired in the MDP.
- We conduct extensive study in the Arcade Learning Environment 100K benchmark with the state-of-the-art algorithms and demonstrate that our proposed method achieves significant performance improvement.
- We show the efficacy of our proposed MaxMin Novelty method in terms of sample efficiency. Our method based on maximizing novelty via minimizing the state-action value function reaches approximately to the same performance level as model-based deep reinforcement learning algorithms, without building and learning any model of the environment.

## 2 BACKGROUND AND PRELIMINARIES

### 2.1 DEEP REINFORCEMENT LEARNING

The reinforcement learning problem is formalized as a Markov Decision Process (MDP) $\mathcal{M} = \langle \mathcal{S}, \mathcal{A}, r, \gamma, \rho_0, \mathcal{P} \rangle$ that contains a continous set of states $s \in \mathcal{S}$, a set of discrete actions $a \in \mathcal{A}$, a probability transition function $\mathcal{T}(s, a, s')$ on $\mathcal{S} \times \mathcal{A} \times \mathcal{S}$, discount factor $\gamma$, a reward function $r(s, a) : \mathcal{S} \times \mathcal{A} \to \mathbb{R}$ with initial state distribution $\rho_0$. A policy $\pi(s, a) : \mathcal{S} \to \mathcal{P}(\mathcal{A})$ in an MDP is a mapping function between states and actions assigning a probability distribution over actions for each state $s \in \mathcal{S}$. The main goal in reinforcement learning is to learn an optimal policy $\pi$ that maximizes the discounted expected cumulative discounted rewards.

$$\mathcal{R} = \mathbb{E}_{a_t \sim \pi(s_t, \cdot)} \sum_t \gamma^t r(s_t, a_t),$$

where $a_t \sim \pi(s_t, \cdot)$. In $Q$-learning the learned policy is parameterized by a state-action value function $Q : \mathcal{S} \times \mathcal{A} \to \mathbb{R}$, which represents the value of taking action $a$ in state $s$. The optimal state-action value function is learnt via iterative Bellman update

$$Q(s_t, a_t) = r(s_t, a_t) + \gamma \sum_{s_t} \mathcal{T}(s_t, a_t, s_{t+1}) \mathcal{V}(s_{t+1}).$$

where $\mathcal{V}(s_{t+1}) = \max_a Q(s_{t+1}, a)$. Let $a^*$ be the action maximizing the state-action value function, $a^*(s) = \arg\max_a Q(s, a)$, in state $s$. Once the $Q$-function is learnt the policy is determined via taking action $a^*(s) = \arg\max_a Q(s, a)$. In deep reinforcement learning, the state space or the action space is large enough that it is not possible to learn and store the state-action values in a tabular form. Thus, the $Q$-function is approximated via deep neural networks.

$$\theta_{t+1} = \theta_t + \alpha(r(s_t, a_t) + \gamma Q(s_{t+1}, \arg\max_a Q(s_{t+1}, a; \theta_t); \theta_t) - Q(s_t, a_t; \theta_t)) \nabla_{\theta_t} Q(s_t, a_t; \theta_t)$$

In deep double-$Q$ learning, two $Q$-networks are used to decouple the $Q$-network deciding which action to take and the $Q$-network to evaluate the action taken.

$$\theta_{t+1} = \theta_t + \alpha(r(s_t, a_t) + \gamma Q(s_{t+1}, \arg\max_a Q(s_{t+1}, a; \theta_t); \hat{\theta}_t) - Q(s_t, a_t; \theta_t)) \nabla_{\theta_t} Q(s_t, a_t; \theta_t)$$

Current deep reinforcement learning algorithms use $\epsilon$-greedy exploration during training (Mnih et al., 2015; Hasselt et al., 2016; Wang et al., 2016; Hamrick et al., 2020; Flennerhag et al., 2022). In particular, the $\epsilon$-greedy algorithm takes an action $a_k \sim \mathcal{U}(\mathcal{A})$ with probability $\epsilon$ in a given state $s$, i.e. $\pi(s, a_k) = \frac{\epsilon}{|\mathcal{A}|}$, and takes an action $a^* = \arg\max_a Q(s, a)$ with probability $1 - \epsilon$, i.e. $\pi(s, a^*) = 1 - \epsilon + \frac{\epsilon}{|\mathcal{A}|}$.

## 2.2 Exploration in Reinforcement Learning

In the tabular MDP setting, there has been extensive theoretical work proving optimal regret bounds using the principal of optimism in the face of uncertainty. One prominent class of algorithms in this setting utilizes a bonus to value estimates based on the Upper Confidence Bound (UCB) approach (Auer et al., 2008). In fact, the recent work of Azar et al. (2017) achieves minimax optimal regret using a carefully designed variant of the UCB approach. Furthermore, the UCB approach to exploration continues to be an active area of research for deriving new algorithms with provable regret bounds in reinforcement learning (Zanette & Brunskill, 2019; Jin et al., 2020). The basic idea of UCB algorithms is to explore by adding an optimistic bonus to the state-action values, based on an estimate of the uncertainty in the current value function. The basic UCB approach (Sutton & Barto, 2018) is to use visit-count statistics $N_t(s, a)$ representing the number of times action $a$ has been taken in state $s$ by time step $t$ in order to estimate the variance of the current state-action values. The variance estimate is then used to construct a confidence interval around the current value estimate, usually given by some multiple of the standard deviation $c\sqrt{\frac{\log t}{N_t(s,a)}}$. Finally, the action with the highest value for the upper end of its confidence interval is selected. In this sense the UCB algorithm is optimistic, as it chooses an action based on the highest plausible estimate of its value given the previously observed data. Note also that as the state action pair $(s, a)$ is visited more frequently, the corresponding confidence interval becomes smaller, eventually converging to the final estimated value. A second general class of theoretically-justified algorithms for exploration is based on randomized value functions, where specifically tuned randomness is added to value estimates in order to encourage exploration. Notable examples of algorithms in this category include Thompson sampling (Osband et al., 2013; Agrawal & Jia, 2017), based on sampling from a posterior distribution on actions given past observations, and randomized least-squares value iteration (RLSVI) (Osband et al., 2016), based on using tuned Gaussian noise to sample a randomized value function.

Despite the strong theoretical performance of the aforementioned approaches, there are significant difficulties in effectively extending to the setting of deep reinforcement learning. The primary obstacle is that these methods utilize count-based uncertainty estimates (e.g. the state-action visit counts $N_t(s, a)$), which are generally not immediately available in deep reinforcement learning where the state space is modeled as a continuous high-dimensional vector space (e.g. in deep reinforcement learning from pixels). Instead, incorporating count-based methods into deep reinforcement learning requires significant complexity including training additional deep neural networks to estimate counts or other uncertainty metrics. As a result, many state-of-the-art deep reinforcement learning algorithms use simple, randomized exploration methods such as the $\epsilon$-greedy approach of sampling a uniformly random action with probability $\epsilon$ (Mnih et al., 2015; Hasselt et al., 2016; Wang et al., 2016; Hamrick et al., 2020; Flennerhag et al., 2022), or the injection of random noise via noisy-networks (Hessel et al., 2018).

## 3 Maximizing Novelty

In deep reinforcement learning the state-action value function is initialized with random weights (Mnih et al., 2015; 2016; Hasselt et al., 2016; Wang et al., 2016; Schaul et al., 2016; Oh et al., 2020; Schrittwieser et al., 2020; Hubert et al., 2021). Thus, in the early phase of the training the $Q$-function will behave more like a random function rather than providing an accurate representation of the optimal state-action values. In particular, early in training the $Q$-function, on average, will assign approximately similar values to states that are similar, and will have little correlation with the immediate rewards. We first formalize this intuition in the following definitions.

**Definition 3.1** ($\eta$-uninformed $Q$). Let $\eta > 0$. A $Q$-function parameterized by weights $\theta \sim \Theta$ is $\eta$-uninformed if for any state $s \in \mathcal{S}$ with $a^{\min} = \arg\min_a Q_\theta(s, a)$ we have

$$|\mathbb{E}_{\theta \sim \Theta}[r(s_t, a^{\min})] - \mathbb{E}_{a \sim \mathcal{U}(\mathcal{A})}[r(s_t, a)]| < \eta.$$

**Definition 3.2** ($\delta$-smooth $Q$). Let $\delta > 0$. A $Q$-function parameterized by weights $\theta \sim \Theta$ is $\delta$-smooth if for any state $s \in \mathcal{S}$ and action $\hat{a} \in \mathcal{A}$ with $s' \sim \mathcal{T}(s, \hat{a}, \cdot)$ we have

$$|\mathbb{E}_{s' \sim \mathcal{T}(s, \hat{a}, \cdot), \theta \sim \Theta}[\max_a Q_\theta(s, a)] - \mathbb{E}_{s' \sim \mathcal{T}(s, \hat{a}, \cdot), \theta \sim \Theta}[\max_a Q_\theta(s', a)]| < \delta$$

where the expectation is over both the random initialization of the $Q$-function weights, and the random transition to state $s' \sim \mathcal{T}(s, \hat{a}, \cdot)$.

**Definition 3.3** (Disadvantage Gap). For a state-action value function $Q_\theta$ the disadvantage gap in a state $s \in \mathcal{S}$ is given by

$$\mathcal{D}(s) = \mathbb{E}_{a \sim \mathcal{U}(\mathcal{A}), \theta \sim \Theta}[Q_\theta(s, a) - Q_\theta(s, a^{\min})]$$

where $a^{\min} = \arg\min_a Q_\theta(s, a)$.

The following proposition captures the intuition that when the $Q$-function on average assigns similar maximum values to consecutive states, choosing the action minimizing the state-action value function will achieve an above-average temporal difference loss.

**Proposition 3.1.** *Let $\eta, \delta > 0$ and suppose that $Q_\theta(s, a)$ is $\eta$-uninformed and $\delta$-smooth. Let $s_t \in \mathcal{S}$ be a state, and let $a^{min}$ be the action minimizing the state-action value in a given state $s_t$, $a^{min} = \arg\min_a Q_\theta(s_t, a)$. Let $s_{t+1}^{min} \sim \mathcal{T}(s_t, a^{min}, \cdot)$. Then for an action $a_t \sim \mathcal{U}(\mathcal{A})$ with $s_{t+1} \sim \mathcal{T}(s_t, a_t, \cdot)$ we have*

$$\mathbb{E}_{s_{t+1}^{min} \sim \mathcal{T}(s_t, a^{min}, \cdot), \theta \sim \Theta}[r(s_t, a^{min}) + \gamma \max_a Q_\theta(s_{t+1}^{min}, a) - Q_\theta(s_t, a^{min})] >$$

$$\mathbb{E}_{a_t \sim \mathcal{U}, (\mathcal{A}) s_{t+1} \sim \mathcal{T}(s_t, a_t, \cdot), \theta \sim \Theta}[r(s_t, a_t) + \gamma \max_a Q_\theta(s_{t+1}, a) - Q_\theta(s_t, a_t)]$$

$$+ \mathcal{D}(s) - 2\delta - \eta$$

*Proof.* Since $Q_\theta(s, a)$ is $\delta$-smooth we have

$$\mathbb{E}_{s_{t+1}^{min} \sim \mathcal{T}(s_t, a^{min}, \cdot), \theta \sim \Theta}[\gamma \max_a Q_\theta(s_{t+1}^{min}, a) - Q_\theta(s_t, a_{min})]$$

$$> \gamma \mathbb{E}_{\theta \sim \Theta}[\max_a Q_\theta(s_t, a)] - \delta - \mathbb{E}_{\theta \sim \Theta}[Q_\theta(s_t, a_{min})]$$

$$> \gamma \mathbb{E}_{s_{t+1} \sim \mathcal{T}(s_t, a_t, \cdot), \theta \sim \Theta}[\max_a Q_\theta(s_{t+1}, a)] - 2\delta - \mathbb{E}_{\theta \sim \Theta}[Q_\theta(s_t, a_{min})]$$

$$\geq \mathbb{E}_{a_t \sim \mathcal{U}(\mathcal{A}), s_{t+1} \sim \mathcal{T}(s_t, a_t, \cdot), \theta \sim \Theta}[\gamma \max_a Q_\theta(s_{t+1}, a) - Q_\theta(s_t, a_t)] + \mathcal{D}(s) - 2\delta$$

where the last line follows from Definition 3.3. Further, because $Q_\theta(s, a)$ is $\eta$-uninformed,

$$\mathbb{E}_{\theta \sim \Theta}[r(s_t, a^{\min})] > \mathbb{E}_{a_t \sim \mathcal{U}(\mathcal{A})}[r(s_t, a_t)] - \eta.$$

Combining with the previous inequality completes the proof. $\square$

In words, the proposition shows that the temporal difference loss achieved by the minimum-value action is above-average by an amount approximately equal to the disadvantage gap.

The above argument can be extended to the case where action selection and evaluation in the temporal difference loss are computed with two different sets of weights $\theta$ and $\hat{\theta}$ as in double $Q$-learning.

**Definition 3.4** ($\delta$-smoothness for Double-$Q$). Let $\delta > 0$. A pair of $Q$-functions parameterized by weights $\theta \sim \Theta$ and $\hat{\theta} \sim \Theta$ are $\delta$-smooth if for any state $s \in \mathcal{S}$ and action $\hat{a} \in \mathcal{A}$ with $s' \sim \mathcal{T}(s, \hat{a}, \cdot)$ we have

$$|\mathbb{E}_{s' \sim \mathcal{T}(s, \hat{a}, \cdot), \theta \sim \Theta, \hat{\theta} \sim \Theta}\left[Q_{\hat{\theta}}(s, \arg\max_a Q_\theta(s, a))\right]$$

$$- \mathbb{E}_{s' \sim \mathcal{T}(s, \hat{a}, \cdot), \theta \sim \Theta, \hat{\theta} \sim \Theta}\left[Q_{\hat{\theta}}(s', \arg\max_a Q_\theta(s', a))\right]| < \delta$$

where the expectation is over both the random initialization of the $Q$-function weights $\theta$ and $\hat{\theta}$, and the random transition to state $s' \sim \mathcal{T}(s, \hat{a}, \cdot)$.

With this definition we can then prove that choosing the minimum valued action will lead to a temporal difference loss that is above-average by approximately $\mathcal{D}(s)$.

**Proposition 3.2.** *Let $\eta, \delta > 0$ and suppose that $Q_\theta$ and $Q_{\hat\theta}$ are $\eta$-uniformed and $\delta$-smooth. Let $s_t \in \mathcal{S}$ be a state, and let $a^{min} = \arg\min_a Q_\theta(s_t, a)$. Let $s_{t+1}^{min} \sim \mathcal{T}(s_t, a^{min}, \cdot)$. Then for an action $a_t \sim \mathcal{U}(\mathcal{A})$ with $s_{t+1} \sim \mathcal{T}(s_t, a_t, \cdot)$ we have*

$$\mathbb{E}_{s_{t+1} \sim \mathcal{T}(s,a,\cdot), \theta \sim \Theta, \hat\theta \sim \Theta}[r(s_t, a^{min}) + \gamma Q_{\hat\theta}(s_{t+1}^{min}, \arg\max_a Q_\theta(s_{t+1}^{min}, a)) - Q_\theta(s_t, a^{min})]$$

$$> \mathbb{E}_{a_t \sim \mathcal{U}(\mathcal{A}), s_{t+1} \sim \mathcal{T}(s,a,\cdot), \theta \sim \Theta, \hat\theta \sim \Theta}[r(s_t, a_t) + \gamma Q_{\hat\theta}(s_{t+1}, \arg\max_a Q_\theta(s_{t+1}, a)) - Q_\theta(s_t, a_t)]$$
$$+ \mathcal{D}(s) - 2\delta - \eta$$

*Proof.* Since $Q_\theta$ and $Q_{\hat\theta}$ are $\delta$-smooth we have

$$\mathbb{E}_{s_{t+1}^{min} \sim \mathcal{T}(s_t, a^{min}, \cdot), \theta \sim \Theta, \hat\theta \sim \Theta}[+\gamma Q_{\hat\theta}(s_{t+1}^{min}, \arg\max_a Q_\theta(s_{t+1}^{min}, a)) - Q_\theta(s_t, a^{min})]$$

$$> \mathbb{E}_{s_{t+1}^{min} \sim \mathcal{T}(s_t, a^{min}, \cdot), \theta \sim \Theta, \hat\theta \sim \Theta}[+\gamma Q_{\hat\theta}(s_t, \arg\max_a Q_\theta(s_t, a)) - Q_\theta(s_t, a^{min})] - \delta$$

$$> \mathbb{E}_{s_{t+1} \sim \mathcal{T}(s_t, a_t, \cdot), \theta \sim \Theta, \hat\theta \sim \Theta}[+\gamma Q_{\hat\theta}(s_{t+1}, \arg\max_a Q_\theta(s_{t+1}, a)) - Q_\theta(s_t, a^{min})] - 2\delta$$

$$\geq \mathbb{E}_{s_{t+1} \sim \mathcal{T}(s_t, a_t, \cdot), \theta \sim \Theta, \hat\theta \sim \Theta}[+\gamma Q_{\hat\theta}(s_{t+1}, \arg\max_a Q_\theta(s_{t+1}, a)) - Q_\theta(s_t, a_t)] + \mathcal{D}(s) - 2\delta$$

where the last line follows from Definition 3.3. Further, because $Q_\theta$ and $Q_{\hat\theta}$ are $\eta$-uniformed,

$$\mathbb{E}_{\theta \sim \Theta, \hat\theta \sim \Theta}[r(s_t, a^{min})] > \mathbb{E}_{a_t \sim \mathcal{U}(\mathcal{A})}[r(s_t, a_t)] - \eta.$$

Combining with the previous inequality completes the proof. $\qquad\square$

At first, the results in Proposition 3.1 and 3.2 might appear counterintuitive. The fact that the $Q$-function is $\delta$-smooth and $\eta$-uninformed seem like properties of a random function. Thus, taking the minimum $Q$-value action should be approximately equivalent to taking a uniform random action. However, Proposition 3.1 and 3.2 show that the temporal difference loss achieved by taking the minimum action is larger than that of a random action by an amount equal to the disadvantage gap $\mathcal{D}(s)$. In order to reconcile these two statements it is useful at this point to look at the limiting case of the $Q$ function at initialization. In particular, the following proposition shows that, at initialization, the distribution of the minimum value action in a given state is uniform by itself, but is constant once we condition on the weights $\theta$.

**Proposition 3.3.** *Let $\theta$ be the random initial weights for the Q-function. For any state $s \in \mathcal{S}$ let $a^{min}(s) = \arg\min_{a' \in \mathcal{A}} Q_\theta(s, a')$. Then for any $a \in \mathcal{A}$*

$$\mathbb{P}_{\theta \sim \Theta}\left[\arg\min_{a' \in \mathcal{A}} Q_\theta(s, a') = a\right] = \frac{1}{|\mathcal{A}|}$$

*i.e. the distribution $\mathbb{P}_{\theta \sim \Theta}[a^{min}(s)]$ is uniform. Simultaneously, the conditional distribution $\mathbb{P}_{\theta \sim \Theta}[a^{min}(s) \mid \theta]$ is constant.*

*Proof.* Since $Q_\theta(s, \cdot)$ is a random function (given the random choice of $\theta$), each action $a \in A$ is equally likely to be assigned the minimum $Q$-value in state $s$. Thus,

$$\mathbb{P}_{\theta \sim \Theta}\left[\arg\min_{a' \in \mathcal{A}} Q_\theta(s, a) = a\right] = \frac{1}{|\mathcal{A}|}.$$

However, given the value of $\theta$, the value of $a^{min}(s)$ is uniquely determined because

$$a^{min}(s) = \arg\min_{a \in \mathcal{A}} Q_\theta(s, a).$$

Therefore, the distribution of $a^{min}(s)$ conditional on $\theta$ is constant. $\qquad\square$

---

**Algorithm 1:** MaxMin Novelty

---

**Input:** In MDP $\mathcal{M}$ with $\gamma \in (0, 1]$, $s \in \mathcal{S}$, $a \in \mathcal{A}$ with $Q(s, a)$, $\mathcal{B}$ experience replay buffer, $\epsilon$ exploration parameter, $\mathcal{N}$ is the training learning steps.

  Populating Experience Replay Buffer:        Learning:

  **for** $s_t$ in $e$ **do**                             **for** $n$ in $\mathcal{N}$ **do**

     Sample $\kappa \sim U(0, 1)$                    Update with probability $\epsilon$:

     **if** $\kappa < \epsilon$ **then**                    $\mathcal{TD} = r(s_t, a^{min})$

       $a^{min} = \arg\min_a Q(s_t, a)$          $+ \gamma \max_a Q(s^{min}_{t+1}, a) - Q(s_t, a^{min})$

       $\mathcal{B} \leftarrow (r(s_t, a^{min}), s_t, s^{min}_{t+1}, a^{min})$     Update with probability $1 - \epsilon$:

     **else**                               $\mathcal{TD} = r(s_t, a^*)$

       $a^* = \arg\max_a Q(s_t, a)$            $+ \gamma \max_a Q(s_{t+1}, a) - Q(s_t, a^*)$

       $\mathcal{B} \leftarrow (r(s_t, a^*), s_t, s_{t+1}, a^*)$      **end for**

     **end if**                          **return** $\nabla \mathcal{L}(\mathcal{TD})$

  **end for**

---

This implies that, in states whose $Q$-values have not changed much from initialization, taking the minimum action is almost equivalent to taking a random action. However, while the action chosen early on in training is almost uniformly random when only considering the current state, it is at the same time completely determined by the current value of the weights $\theta$. The temporal difference loss is also determined by the weights $\theta$. Thus while the marginal distribution on actions taken is uniform, the temporal difference loss when taking the minimum action is quite different than from the case where an independently random action is chosen.

Algorithm 1 summarizes our proposed exploration method MaxMin Novelty based on minimizing the state-action value function as described in detail in Section 3. Note that populating the experience replay buffer and learning are happening simultaneously with different rates.

## 4  MOTIVATING EXAMPLE

As a motivating example we consider the chain MDP which consists of a chain of $n$ states $s \in \mathcal{S} = \{1, 2, \cdots n\}$ each with two actions. Each state $i$ has one action that transitions the agent up the chain by one step to state $i + 1$, and one action which resets the agent to state 1 at the beginning of the chain. All transitions have reward zero, except for the last transition returning the agent to the beginning from the $n$-th state. Thus, when started from the first state in the chain, the agent must learn a policy that takes $n - 1$ consecutive steps up the chain, and then the one final step to reset and get the reward.

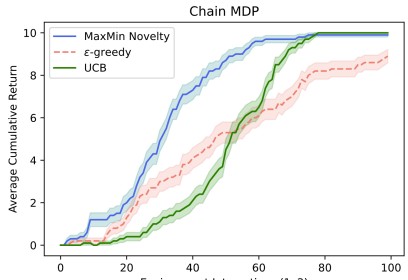

Figure 1: Exploring the chain MDP with Upper Confidence Bound (UCB) method, $\epsilon$-greedy and our proposed method MaxMin Novelty.

For the chain MDP, we compare standard approaches to exploration in tabular $Q$-learning with our method MaxMin Novelty based on minimization of the state-action values. In particular we compare our method MaxMin Novelty with both the $\epsilon$-greedy action selection method, and the upper confidence bound (UCB) method. In more detail, in the UCB method the number of training steps $t$, and the number of times $N_t(s, a)$ that each action $a$ has been taken in state $s$ by step $t$ are recorded. Furthermore, the action $a \in \mathcal{A}$ selection is determined as follows:

$$a^{\text{UCB}} = \arg\max_{a \in \mathcal{A}} Q(s, a) + 2\sqrt{\frac{\log t}{N_t(s, a)}}.$$

In a given state $s$ if $N(s, a) = 0$ for any action $a$, then an action is sampled uniformly at random from the set of actions $a'$ with $N(s, a') = 0$. For the experiments reported in our paper the length of the chain is set to $n = 10$, and $\epsilon = 0.2$. The $Q$-function is initialized by independently sampling each state-action value from a normal distribution with $\mu = 0$ and $\sigma = 0.1$. In each iteration we

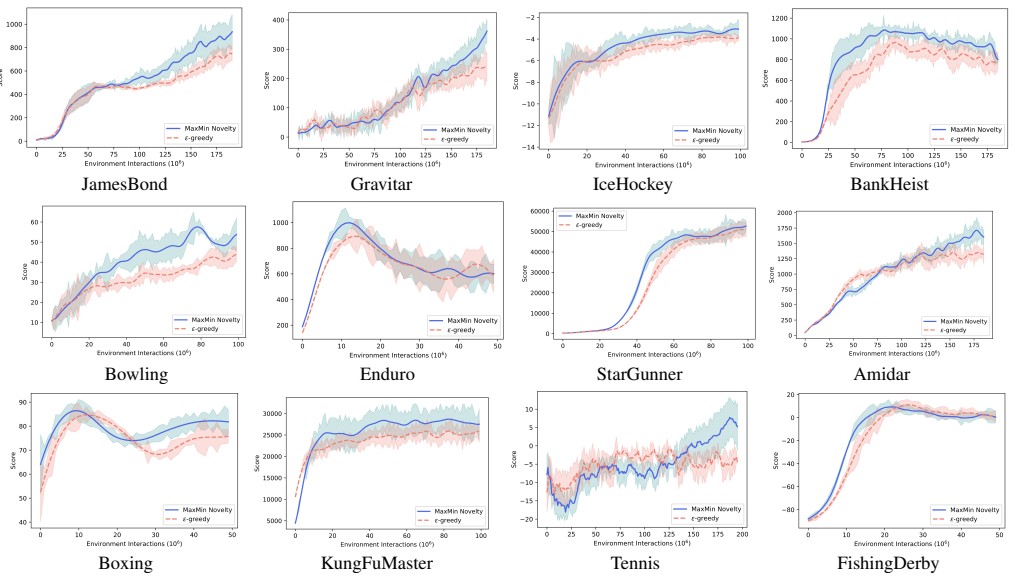

Figure 2: The learning curves of StarGunner, FishingDerby, Boxing, Enduro, Bowling, IceHockey, BankHeist, JamesBond, KungFuMaster, Amidar, Gravitar and Tennis with our proposed method MaxMin Novelty and the $\epsilon$-greedy algorithm in the Arcade Learning Environment with 200 million frame training.

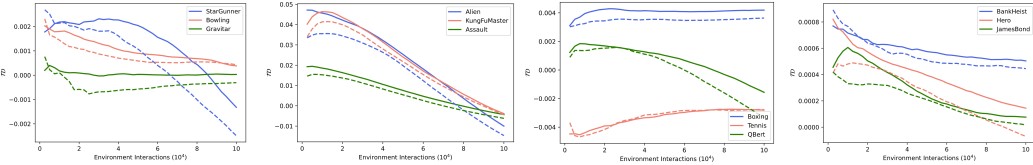

Figure 3: Temporal difference loss for our proposed algorithm MaxMin-Novelty and the canonical $\epsilon$-greedy algorithm in the Arcade Learning Environment 100K benchmark. Dashed lines report the temporal difference loss for the $\epsilon$-greedy algorithm and solid lines report the temporal difference loss for the MaxMin-Novelty algorithm. Colors indicate games.

train the agent using $Q$-learning for 100 steps, and then evaluate the reward obtained by the argmax policy using the current $Q$-function for 100 steps. Note that the maximum achievable reward in 100 steps is 10. The learning curves in Figure 1 demonstrate that our method converges more quickly to the optimal policy than either of the standard approaches.

## 5 LARGE SCALE EXPERIMENTAL RESULTS

The experiments are conducted in the Arcade Learning Environment (ALE) (Bellemare et al., 2013). The Double-$Q$ Network (Hasselt et al., 2016) initially proposed by (van Hasselt, 2010) is trained with prioritized experience replay (Schaul et al., 2016) without the dueling architecture with its original version (Hasselt et al., 2016). The experiments are conducted both in the 100K Arcade Learning Environment benchmark (van Hasselt et al., 2019), and the canonical version with 200 million frame training. Note that the 100K Arcade Learning Environment benchmark is an established baseline proposed to measure sample efficiency in deep reinforcement learning research. The ALE 100K benchmark contains 26 different Arcade Learning Environment games. The policies are evaluated after 100000 environment interactions. All of the polices in the experiments are trained over 5 random seeds. The hyperparameters and the architecture details are reported in the appendix. All of the results in the paper are reported with the standard error of the mean. The human normalized scores are computed as,

$$\text{HN} = \frac{\text{Score}_{agent} - \text{Score}_{random}}{\text{Score}_{human} - \text{Score}_{random}} \quad (1)$$

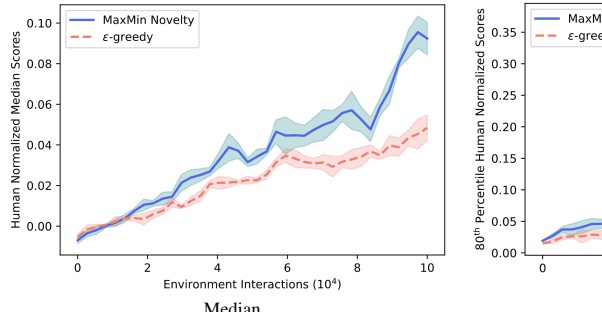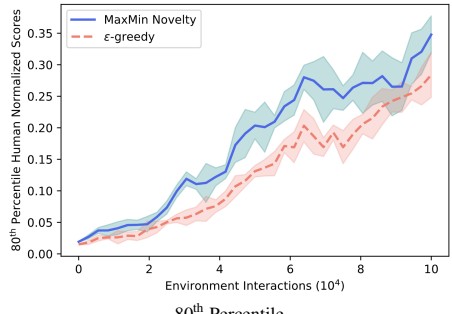

Median                    80[th] Percentile

Figure 4: Human normalized scores median and 80[th] percentile over all games in the Arcade Learning Environment (ALE) 100K benchmark for MaxMin Novelty algorithm and the canonical exploration algorithm $\epsilon$-greedy.

Table 1: Human normalized scores median and 20[th] percentile across all of the games in the Arcade Learning Environment 100K benchmark for MaxMin-Novelty, $\epsilon$-greedy and NoisyNetworks.

| Method | Human Normalized Median | 20[th] Percentile | 80[th] Percentile |
|---|---|---|---|
| MaxMin-Novelty | **0.0927±0.0050** | **0.0145±0.0003** | **0.3762±0.0137** |
| $\epsilon$-greedy | 0.0377±0.0031 | 0.0056±0.0017 | 0.2942±0.0233 |
| NoisyNetworks | 0.0457±0.0035 | 0.0102±0.0018 | 0.1913±0.0144 |

For completeness we also report several results with 200 million frame training (i.e. 50 million environment interactions). In particular, Figure 2 demonstrates the learning curves for our proposed algorithm MaxMin Novelty and the original version of the DDQN algorithm with $\epsilon$-greedy training (Hasselt et al., 2016). In the large data regime we observe that while in some MDPs our proposed method MaxMin Novelty based on exploring with novelty maximization via minimizing the state-action values converges faster, in other MDPs MaxMin Novelty simply converges to a better policy. More concretely, while the learning curves of StarGunner, FishingDerby, Boxing, Enduro, Hero, and IceHockey games in Figure 2 demonstrate the faster convergence rate of our proposed algorithm MaxMin Novelty, the learning curves of the BankHeist, JamesBond, KungFuMaster, Amidar, Gravitar and Tennis games demonstrate that our exploration technique not only increases the sample efficiency in deep reinforcement learning, but also results in learning a policy that is more close to optimal compared to learning a policy with the original method used in the DDQN algorithm.

Additionally, we also compare our proposed MaxMin Novelty algorithm with NoisyNetworks as described in Section 2.2. Table 1 further demonstrates that the MaxMin Novelty algorithm achieves significantly better performance results compared to NoisyNetworks. Furthermore, note that NoisyNetworks includes adding layers in the $Q$-network to increase exploration. However, this increases the number of parameters that have been added in the training process; thus, introducing additional cost to increase exploration. Table 1 reports results of human normalized median scores, 20[th] percentile, and 80[th] percentile for the Arcade Learning Environment 100K benchmark. Thus, Table 1 demonstrates that our proposed MaxMin-Novelty algorithm improves on the performance of the canonical algorithm $\epsilon$-greedy by 248% and NoisyNetworks by 204%.

## 6   INVESTIGATING THE TEMPORAL DIFFERENCE LOSS

The original justification for exploring with the minimum $Q$-value action, is that taking this action tends to result in transitions with higher temporal difference loss. The theoretical analysis from Proposition 3.1 indicates that, when the $Q$ function is $\delta$-smooth and $\eta$-uninformed, taking the minimum value action results in an increase in the temporal difference loss proportional to the disadvantage gap. In particular, Proposition 3.1 states that the temporal difference loss achieved when taking the minimum $Q$-value action in state $s$ exceeds the average loss over a uniform random action by $\mathcal{D}(s) - 2\delta - \eta$.

In order to evaluate how well the theoretical prediction matches reality, in this section we provide empirical measurements of the temporal difference loss in our experiments. To measure the change in the loss when taking the minimum action versus the average action, we compare the temporal

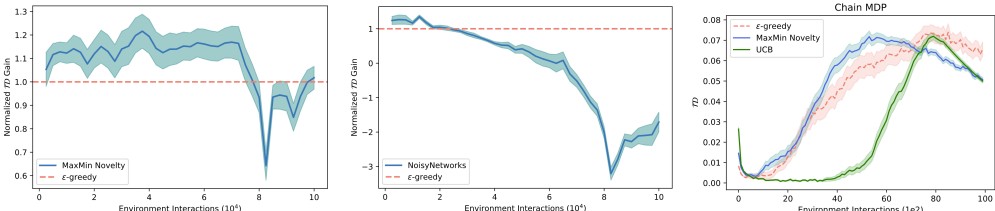

Figure 5: Left and Middle: Normalized temporal difference $\mathcal{TD}$ gain median across all games in the Arcade Learning Environment 100K benchmark for MaxMin Novelty and NoisyNetworks. Right: Temporal difference loss $\mathcal{TD}$ when exploring chain MDP with Upper Confidence Bound (UCB) method, $\epsilon$-greedy and our proposed algorithm MaxMin Novelty.

difference loss obtained by MaxMin Novelty exploration with that obtained by $\epsilon$-greedy exploration. In more detail, during training, for each batch $\Lambda$ of transitions of the form $(s_t, a_t, s_{t+1})$ we record, the temporal difference loss

$$\mathcal{TD} = \mathbb{E}_{(s_t,a_t,s_{t+1})\sim\Lambda}\mathcal{TD}(s_t, a_t, s_{t+1})$$
$$= \mathbb{E}_{(s_t,a_t,s_{t+1})\sim\Lambda}[r(s_t, a_t) + \gamma \max_a Q_\theta(s_{t+1}, a) - Q_\theta(s_t, a_t)].$$

The results reported in Figure 3 and Figure 5 further confirm the theoretical predictions made via Definition 3.2 and Proposition 3.1. In addition to the results for individual games reported in Figure 3, we compute a normalized measure of the gain in temporal difference achieved when using MaxMin Novelty exploration and plot the median across games. We define the normalized $\mathcal{TD}$ gain to be

$$\text{Normalized } \mathcal{TD} \text{ Gain} = 1 + \frac{\mathcal{TD}_{\text{method}} - \mathcal{TD}_{\epsilon\text{-greedy}}}{|\mathcal{TD}_{\epsilon\text{-greedy}}|}$$

where $\mathcal{TD}_{\text{method}}$ and $\mathcal{TD}_{\epsilon\text{-greedy}}$ are the temporal difference for any given exploration method and $\epsilon$-greedy respectively. The leftmost and middle plot of Figure 5 report the median across all games of the normalized $\mathcal{TD}$ gain results for MaxMin Novelty and NoisyNetworks in the Arcade Learning Environment 100K benchmark. Note that, consistent with the predictions of Proposition 3.1, the median normalized temporal difference gain for MaxMin Novelty is up to 25 percent larger than that of $\epsilon$-greedy. The results for NoisyNetworks demonstrate that alternate exploration methods lack this positive bias relative to the uniform random action. The fact that, as demonstrated in Table 1, MaxMin Novelty significantly outperforms noisy networks in the low-data regime is further evidence of the advantage the positive bias in temporal difference confers. The rightmost plot of Figure 5 reports $\mathcal{TD}$ for the motivating example of the chain MDP. As in the large-scale experiments, prior to convergence MaxMin Novelty exhibits a notably larger temporal difference loss relative to the other exploration methods.

## 7 CONCLUSION

In our study we focus on the following questions in deep reinforcement learning: *(i) Is it possible to increase sample efficiency in deep reinforcement learning in a computationally efficient way with conceptually simple choices?, (ii) What is the theoretical motivation of our proposed perspective, simply minimizing the state-action value function in early training, that results in one of the most computational efficient ways to explore in deep reinforcement learning?* and, *(iii) How would the theoretically motivated simple idea transfer to large scale experiments in high-dimensional state representation MDPs?* To be able to answer these questions we propose a novel, theoretically motivated method with zero additional computational cost based on following actions that minimize the state-action value function to explore in deep reinforcement learning. We demonstrate theoretically that our method MaxMin Novelty based on minimization of the state-action value results in higher temporal difference loss, and thus creates novel transitions in exploration with more unique experience collection. Following the theoretical motivation we initially show in a toy example in the chain MDP setup that our proposed method MaxMin Novelty results in achieving higher sample efficiency. Then, we expand this intuition and conduct large scale experiments in the Arcade Learning Environment, and demonstrate that our proposed method MaxMin Novelty increases the performance on the Arcade Learning Environment 100K benchmark by $248\%$.

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

## A  APPENDIX

### A.1  HYPERPARAMETER AND ARCHITECTURE DETAILS

For reproducibility and completeness in research in Table 2 we report the hyperparameter details for our proposed algorithm MaxMin-Novelty, canonical algorithm $\epsilon$-greedy and NoisyNetworks. Furthermore, for all of the algorithms the hyperparameters and the architectures are identical with each other. Note that the architecture parameters are also identical for the 200 million frame training. Note that we did not tune hyperparameters reported below. To increase transparency in research we kept hyperparameters exactly the same with the prior studies. We ran our experiments with JAX implementation Bradbury et al. (2018). We used Haiku Hennigan et al. (2020) for the neural network library, Optax Hessel et al. (2020) for the optimization library, and RLax for the reinforcement learning library Babuschkin et al. (2020).

Table 2: Hyperparameters and architectures used in the experiments for our proposed algorithm MaxMin-Novelty, canonical algorithm $\epsilon$-greedy and NoisyNetworks.

| Hyperparameters | Settings (For all of the algorithms) |
|---|:---:|
| Grey-scaling | True |
| Observation down-sampling | (84, 84) |
| Frames stacked | 4 |
| Action repetitions | 4 |
| Batch Size | 32 |
| Update | Double-Q |
| Max Frames per episode | 108000 |
| Exploration epsilon | 0.01 |
| Evaluation exploration epsilon | 0.01 |
| Exploration epsilon decay frame fraction | 0.008 |
| Gradient error bound | 0.03125 |
| Learning rate | 0.00025 |
| Optimizer epsilon | $0.01/32^2$ |
| Optimizer | Adam |
| Discount factor | 0.99 |
| Maximum absolute rewards | 1 |
| Number of iterations | 40 |
| Number of training frames | $10^4$ |
| Nesterov Momentum | True |
| Hardware | GPU |
| NoisyNetwork parameter | 0.1 |
| $Q$-Network channels | 32,64,64 |
| $Q$-Network filter size | $8 \times 8, 4 \times 4, 3 \times 3$ |
| $Q$-Network stride | $(4, 4), (2, 2), (1, 1)$ |
| $Q$-Network hidden units | 512 |

## A.2  ARCADE LEARNING ENVIRONMENT RESULTS

Table 3 reports the average scores for human, random, our proposed algorithm MaxMin-Novelty, canonical algorithm $\epsilon$-greedy and NoisyNetworks for all of the games in the Arcade Learning Environment 100K benchmark. Scores are reported with the mean over 5 random seeds. The highest score amongst the three algorithms is marked with bold font. We also reported human scores and random scores to provide complete information on the learning curves reported in the main body of the paper.

Table 3: Average returns for human, random, our proposed algorithm MaxMin-Novelty, canonical algorithm $\epsilon$-greedy and NoisyNetworks across all of the games in the Arcade Learning Environment 100K benchmark. Scores are averaged over 5 random seeds.

| Games | Human | Random | $\epsilon$-greedy | NoisyNetworks | MaxMin-Novelty |
|---|---|---|---|---|---|
| Alien | 7127.7 | 227.8 | 498.47 | 466.50 | **595.70** |
| Amidar | 1719.5 | 5.8 | 42.31 | **42.83** | 41.94 |
| Assault | 742.0 | 222.4 | **396.00** | 375.72 | 383.25 |
| Asterix | 8503.3 | 210.0 | 306.36 | 305.64 | **410.07** |
| BankHeist | 753.1 | 14.2 | **15.72** | 13.1 | 13.45 |
| BattleZone | 37187.5 | 2360.0 | 1844.61 | 1100.00 | **2200.00** |
| Boxing | 12.1 | 0.1 | 7.25 | 4.6 | **7.9** |
| Breakout | 30.5 | 1.7 | 4.83 | 5.33 | **9.03** |
| ChopperCommand | 7387.8 | 811.0 | 639.48 | 919.83 | **987.83** |
| CrazyClimber | 35829.4 | 10780.5 | 10075.00 | **18550.00** | 9870.00 |
| DemonAttack | 1971.0 | 152.1 | **1365.60** | 576.5 | 745.00 |
| Freeway | 29.6 | 0.0 | 0.00 | 5.1 | **14.00** |
| FrostBite | 4334.7 | 65.2 | 184.41 | 167.60 | **206.40** |
| Gopher | 2412.5 | 257.6 | 633.84 | 468.66 | **664.00** |
| Hero | 30826.4 | 1027.0 | 1628.42 | **1884.50** | 1528.40 |
| Jamesbond | 302.8 | 29.0 | 21.73 | **22.08** | 19.33 |
| Kangaroo | 3035.0 | 52.0 | 251.00 | 90.00 | **280.83** |
| Krull | 2665.5 | 1598.0 | **2206.02** | 2040.50 | 1491.5 |
| KungFuMaster | 22736.3 | 258.5 | **7116.94** | 5665.00 | 4045.00 |
| Mspacman | 6951.6 | 307.3 | 719.16 | **912.83** | 671.08 |
| Pong | 14.6 | -20.7 | -8.18 | **-2.3** | -6.30 |
| PrivateEye | 69571.3 | 24.9 | 0.25 | -7.50 | **30.0** |
| Qbert | 13455.0 | 163.9 | **519.71** | 556.08 | 466.83 |
| RoadRunner | 7845.0 | 11.5 | **3600.85** | 1527.35 | 670.66 |
| Seaquest | 42054.7 | 68.4 | 206.49 | 333.66 | **348.00** |
| UpNdDown | 11693.2 | 533.4 | 1858.41 | 1948.00 | **1953.91** |

Figure 6 reports learning curves in the Arcade Learning Environment 100K benchmark with our proposed algorithm MaxMin Novelty and the canonical algorithm $\epsilon$-greedy. In early training (i.e. up to $4 \times 10^4$ environment interactions) in half of the games we observe a steeper increase in the performance. This is again a result of the MaxMin Novelty algorithm targeting higher temporal difference bias. In particular, in Amidar, CrazyClimber, Hero, JamesBond, Kangaroo, RoadRunner, PrivateEye, Seaquest, UpNDown, Freeway, Breakout and Asterix the gradient of the performance curve for MaxMin Novelty is higher than the canonical algorithm $\epsilon$-greedy. This is again supporting Proposition 3.6 in the main body of the paper. In particular, early in the training the $Q$-function is $\eta$-uninformed and $\delta$-smooth as has been described in Definition 3.1 and 3.2 in the main body of the paper. Thus, the steep increase in early training matches the predictions of Proposition 3.6 where a positive bias in temporal difference yields faster learning.

Also further note that in 6 games[1] simple double-$Q$ learning already outperforms Rainbow in the low data regime. Note that Rainbow has several additional components such as dueling network, multi-step return, distributional reinforcement learning that introduces new parameters as large as

---

[1]Boxing, Breakout, ChopperCommand, DemonAttack, Gopher, Pong

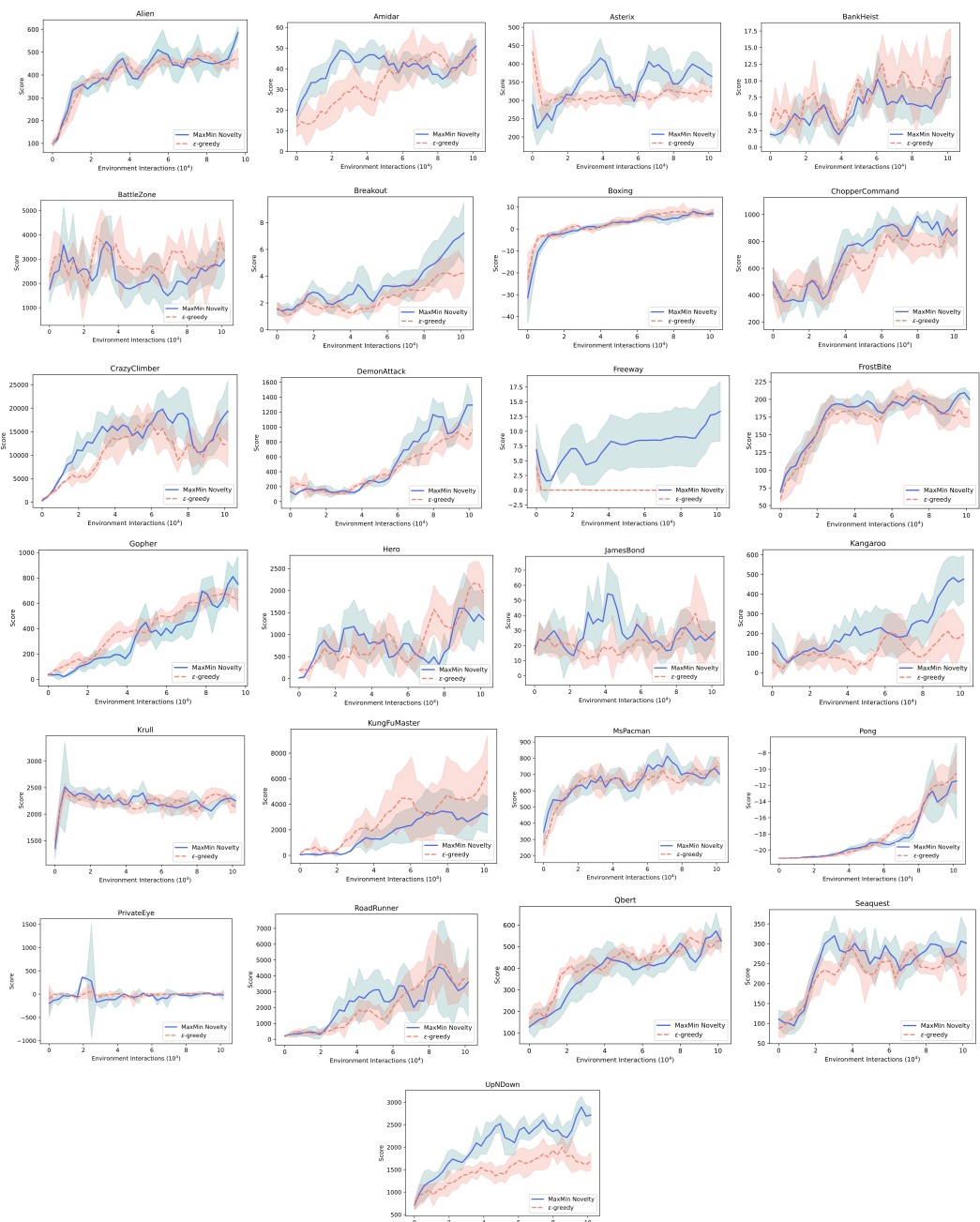

Figure 6: Learning curves in the Arcade Learning Environment 100K benchmark with our proposed algorithm MaxMin Novelty and the canonical algorithm $\epsilon$-greedy.

the number of bins used in the algorithm, and NoisyNetworks. Thus, the fact that MaxMin Novelty with simple double-$Q$ learning achieves a higher score in these games than an algorithm that combines all these various techniques is further evidence that demonstrates the NoisyNetworks can be replaced with the MaxMin Novelty algorithm in Rainbow as a future research direction to obtain better performance. Also further note that MaxMin Novelty does not introduce any additional new parameters as NoisyNetworks does; more precisely, NoisyNetworks doubles the number of parameters used in the $Q$-network. Hence, the fact that MaxMin Novelty achieves higher performance as also reported in the main body of the paper without any additional computational cost further demonstrates the benefits of the utilization of MaxMin Novelty in a more diversified portfolio of algorithms as a zero cost exploration technique.

### A.3 MOTIVATING EXAMPLE RESULTS

In this section we provide more results into the motivating example of the chain-MDP. In particular, while the main body of the paper provides results with the baseline chain-MDP in this section we provide more results with the modified chain-MDP. In detail, the modified chain-MDP refers to the chain-MDP with increased action size to obtain more fine-grained observations into the effects of the exploration techniques. Hence, the modified chain-MDP consists of $n$ states $s \in \mathcal{S} = 1, 2, \dots, n$ each with four actions. In the modified chain-MDP each state $i$ has one action that transitions the agent up the chain by one step to state $i + 1$, one action that transitions the agent to state two, one action that transitions the agent to state three, and one action which resets the agent to state one at the beginning of the chain. The Figure 7 reports results for MaxMin Novelty, canonical $\epsilon$-greedy, and the UCB method with varying $\epsilon \in [0.15, 0.25]$ with a step size of 0.025. The results reported in Figure 7 once more demonstrate that MaxMin Novelty performs significantly better compared to prior exploration techniques.

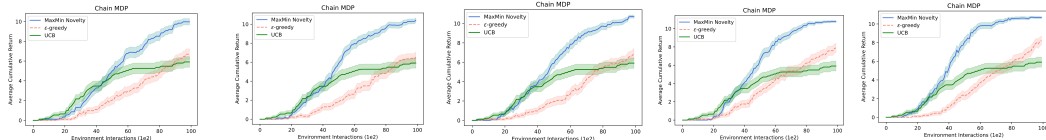

Figure 7: Learning curves in the chain MDP with our proposed algorithm MaxMin Novelty, the canonical algorithm $\epsilon$-greedy and the UCB algorithm with variations in $\epsilon$.

### A.4 EXTENSION OF MAXMIN NOVELTY TO DIFFERENT REINFORCEMENT LEARNING ALGORITHMS

While the main paper focuses on Deep Double $Q$-Network, Figure 8 reports human normalized median and human normalized 80[th] percentile scores for MaxMin Novelty and $\epsilon$-greedy with the QRDQN algorithm[2] across all of the tasks of the Arcade Learning Environment 100K benchmark. With MaxMin Novelty QRDQN is able to achieve 0.15582 human normalized median score. Note that data-efficient Rainbow can only achieve 0.12 human normalized median scores van Hasselt et al. (2019). Furthermore, note that Rainbow contains dueling architecture, multi-step return, noisy networks on top of the distributional reinforcement learning. Thus, the fact that QRDQN, a baseline distributional reinforcement learning algorithm, can achieve human normalized median score that is already substantially higher than data-efficient Rainbow once more demonstrates the substantial sample efficiency gained by the MaxMin Novelty algorithm. Furthermore, the significantly higher performance gain obtained by MaxMin Novelty over all tasks of the Arcade Learning Environment 100K benchmark as reported via the human normalized scores in Figure 8, once more demonstrates that MaxMin Novelty increases the performance of the baseline algorithms further beyond the performance of much more complicated algorithms.

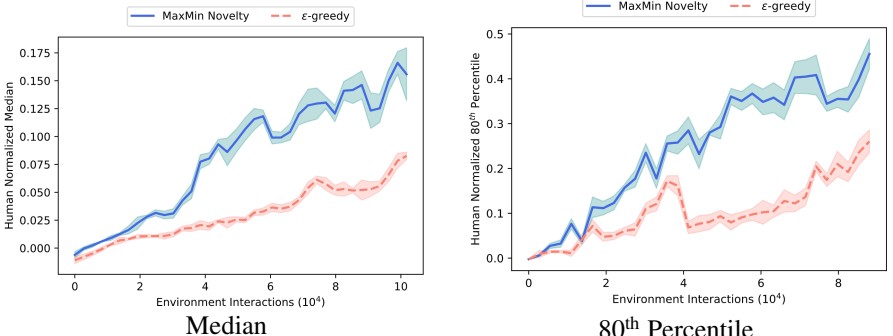

Median      80[th] Percentile

Figure 8: Human normalized median and human normalized 80[th] percentile scores of QRDQN with MaxMin Novelty and $\epsilon$-greedy in the Arcade Learning Environment 100K benchmark.

---

[2]QRDQN is one of the baseline distributional reinforcement learning algorithms.

