# OpenReview forum: "MaxMin-Novelty: Maximizing Novelty via Minimizing the State-Action Values in Deep Reinforcement Learning"
_ICLR.cc/2023/Conference — Submitted to ICLR 2023_

### Official Review · Reviewer_3c9Y · 2022-10-16

**Confidence:** 4
**Correctness:** 2
**Technical Novelty And Significance:** 2
**Empirical Novelty And Significance:** 2
**Recommendation:** 3

**Clarity, Quality, Novelty And Reproducibility:**

CLARITY
The paper has several elements that should be written differently. In the abstract only, here are three elements:
- It is written that deep RL "achieved high acceleration in its progress". A constant progress would already mean that things improve steadily. An accelerating progress is even more specific and I think that using such terms is debatable and should be avoided in a technical abstract.
- It is written that efficient exploration is "a well-understudied and ongoing open problem". Beyond the choice of the word "well-understudied", it feels a bit odd to say that exploration is "understudied" because exploration is known as being one of the top challenges in RL.
- The algorithm is described as coming "with zero computational cost". It should be made clear that it is compared to another algorithm. In addition, the term "zero" is quite precise and I'm unsure you can actually claim this.

Additional elements lack clarity, for instance in the Algorithm 1: What is $e$? It seems that the replay buffer is not used in the learning phase?

QUALITY, NOVELTY
- Conceptually, my main concern is that the strategy of the algorithm does not sound reasonable. Imagine that you have initialized all Q-values at 0 for a given state. Imagine now that from that state, one of the actions gets a negative Q-value (e.g. negative reward followed by terminal state) and another one a postive Q-value (e.g. small positive reward followed by terminal state). In that case, it seems that the algorithm would never explore the other actions in that state.
- The paper claims that it demonstrates "theoretically that our method MaxMin Novelty based on minimization of the state-action value results in higher temporal difference loss, and thus creates novel transitions in exploration with more unique experience collection". This claim is very strong (as it seems to be always true, including with deep learning as a function approximator). That is not what is see in terms of theoretical results in the paper.
- From Table 3 in the appendix, the results are much less convincing as compared to what they are presented in the paper.

REPRODUCIBILITY
The source code does not seem to be provided.

**Strength And Weaknesses:**

Strengths:
The claims of the paper are ambitious.

Weaknesses:
The paper is not sound.

**Summary Of The Paper:**

The paper claims in the abstract that they develop a method for exploration that is "theoretically well motivated, and comes with zero computational cost while leading to significant sample efficiency gains in deep reinforcement learning training". The claim about their experiments is that the "technique improves the human normalized median scores of Arcade Learning Environment by 248% in the low-data regime".

**Summary Of The Review:**

The paper studies a potentially interesting setting but the claims are not well substantiated.

---

> ### Author Response · Authors · 2022-11-07
> **Author Response**
>
> We believe currently your review is not well substantiated for the score you provided. We hope to have a reasonable discussion with you regarding the content of the paper during the response period.
>
> 1. *”From Table 3 in the appendix”*
>
> We believe that you might have some confusion here. Table 3 reports  average returns for human, random, our proposed algorithm MaxMin-Novelty and for $\epsilon$ greedy in the standard form. Table 2 reports human normalized median, human normalized 20th percentile and human normalized 80th percentile scores. Thus, if you compute the human normalized scores by the human scores and random scores as reported in Table 3 in column 2 and 3 with the Equation 1 you would obtain Table 2. Human normalized median score across Arcade Learning Environment games is the main standard metric used in every single deep reinforcement learning algorithm design paper. Can you please be more specific what you refer to here as less convincing?
>
> 2. *”Conceptually, my main concern is “*
>
> First of all, the invalidity of your concern that is based on a simple bandit example is already demonstrated by the extensive experiments provided in the paper.
> We believe that it is perhaps not the best approach to conceptualize deep reinforcement learning as a bandit problem since deep reinforcement learning is much more complicated and complex than that.
> While it is possible to come up with trivial specific examples in bandit problems claiming certain actions will not be explored, these are quite far from the utter reality of how deep reinforcement learning works.
>
> Can you please explain based on what exactly you are claiming that the paper is not sound, and the claims are not well substantiated?
>
> We feel like you have perhaps some strong opinions about some of the phrasing in the abstract and the conclusion that occupies the majority of your review. However, we don’t see this as a barrier for us to further improve the clarity and the phrasing of the paper. We believe the review period serves the purpose of us working together to produce a better version of the presentation of our scientific results.
> Thus, we would be happy to improve the phrasing if this is considered more clear and concise.
>
> Do you have any concrete problems with the actual scientific content of the paper?

---

> > ### Comment · Reviewer_3c9Y · 2022-11-07
> > **Additional details**
> >
> > About my main concern (also raised by Reviewer 7Z6y), an algorithm that is used in the context of deep RL should according to me have some reasonable motivations. Given the simple example of a failure case, I think that the paper currently lacks a clear motivation for the approach. The fact that it is failing in such examples is according to me worrying as it could also happen in a deep RL setting.
> >
> > Given that the method is not grounded on strong theory/motivation, I would then expect that the other elements from the paper develop clearly the advantage of the approach and provides empirical evidence that match the claims, which I don't really feel is the case currently (see other comments). In addition, the writing should also be improved according to my review (see above).
> >
> > To answer your comment concerning Table 3 in the appendix, the scores mention that the approach is better than the baselines ε-greedy and NoisyNetworks on half of the environments (13 out of 26).
> >
> > I have read the other reviews and as of now I keep a reject score, but without a strong reject (score 3).

---

> > > ### Author Response · Authors · 2022-11-09
> > > **Response**
> > >
> > > Thank you for your response. We are glad that we can have a common ground on holding discussions further relevant to the scientific merits of the paper.
> > >
> > > Can you please explain which comments you refer to by “see other comments”?
> > >
> > > The empirical evidence in our paper indeed matches the claims. Can you please explain what you mean here?
> > >
> > > 1. *“Table 3”*
> > >
> > > Table 3 shows that the $\epsilon$-greedy performs better in 7 games, NoisyNetworks performs better in 6 games and MaxMin-Novelty performs the best in 13 games. Thus MaxMin Novelty performs the best in almost double the number of games from the prior algorithms in the Arcade Learning Environment 100K benchmark. Nonetheless, it still might be a more stable method of comparison to look at human normalized median scores since this is the main comparison metric that is being used in deep reinforcement learning algorithm comparison. Thus, again the human normalized median score of MaxMin Novelty is more than double that of the prior methods.
> > >
> > > 2. *“Motivation”*
> > >
> > > The concrete motivation of the proposed method, as has been also discussed in Section 3 in Proposition 3.1 and Proposition 3.2, is that the temporal difference loss obtained by taking the action that minimizes the state-action value function for a given state is higher than the temporal difference loss obtained by taking a random action. On the contrary to your claim **achieving higher temporal difference loss** is a clear strong motivation to achieve high sample efficiency in reinforcement learning.
> > > Can you please explain why you would think the opposite in terms of the relationship between achieving higher temporal difference loss and sample efficiency?
> > >
> > > Regarding the bandit related problem, if there is a concern about a type of bandit relevant counter example this can be circumvented by simply adding some noise to MaxMin Novelty to create a "soft" version where with probability $p\cdot\epsilon$ the minimum action is taken and with probability $(1-p)\cdot\epsilon$ a random action is taken where the probability p is close to 1 ($p \approx 1$). This noise will ensure that all actions have some chance of being taken, ruling out the particular concerns regarding the bandit settings, while putting larger weight on the minimum value action to achieve higher temporal difference loss to increase sample efficiency as described in our paper.
> > > The extensive experiments in our paper demonstrate that the specific example you mention had no impact in deep reinforcement learning. The fact that MaxMin-Novelty yields significantly higher human normalized median scores across all of the tasks in the Arcade Learning Environment is a quite significant signal on this.

---

> ### Author Response · Authors · 2022-11-18
> **A new Section in the Appendix**
>
> We have added a new Section in the appendix dedicated to extending MaxMin Novelty over different reinforcement learning algorithms. The results reported in Section A4 once more demonstrate that MaxMin Novelty results in substantially higher performance than prior canonical exploration techniques; furthermore, MaxMin Novelty enables baseline reinforcement learning algorithms to perform substantially higher beyond much more complicated algorithms that contain multiple additional advanced techniques including multi step return, noisy networks, and dueling architecture.

---

### Official Review · Reviewer_7Z6y · 2022-10-24

**Confidence:** 3
**Clarity, Quality, Novelty And Reproducibility:** The paper was easy to follow. The the…
**Correctness:** 2
**Technical Novelty And Significance:** 2
**Empirical Novelty And Significance:** 2
**Recommendation:** 3

**Strength And Weaknesses:**

Strengths
- I found the chain MDP experiment to be quite interesting to transmit the issue the authors want to tackle. That is, the fixed noise that comes from initialization of parametric Q functions.

Unfortunately, I have some major worries with this paper.

Weaknesses
- After some thought, I realized that the proposed method isn't guaranteed to converge to an optimal policy even in very simple examples. Take for example a 3-armed bandit problem with rewards 0, 1 and 2 per arm. An agent that learnt the Q-values 0, 1 and 0.5 for the arms (in the same order as before), will take the second arm (maximum Q value), except for when it's exploring that it will take the first arm (minimum Q value). Thus, it will never be able to explore the third arm, which is the most rewarding one. This is already a big red flag about this algorithm.
- On top of the main issue with the theory of the paper, I don't find the experimental results convincing. In order to make a fair comparison HP-searches must be ran for each individual algorithm. At the very least, the value of $\epsilon$ should be tuned differently, as it has a very different role in each algorithm. In fact, model-free approaches have been shown to achieve much higher scores in [1]. I suspect the 0.03 of $\epsilon$-greedy from this paper comes mainly from a lack of proper hyper-parameters.

[1] https://arxiv.org/pdf/1906.05243.pdf

**Summary Of The Paper:**

As an alternative to the usual $\epsilon$-greedy exploration used in DQN and similar approaches, the authors propose taking the action with the minimum Q value with probability $\epsilon$.

**Summary Of The Review:**

The proposed approach isn't guaranteed to converge to an optimal policy, even in the tabular case. While this could be acceptable if the experimental part of the paper was very solid, that is not the case either. I expect the authors to clearly state that the approach doesn't converge to the optimal policy if they plan to submit this paper to any conference/journal.

---

> ### Author Response · Authors · 2022-11-08
> **Author Response**
>
> Thank you for devoting your time to provide feedback on our paper. Below we have tried to address your questions.
>
> 1. *”Experimental Results”*
>
> We believe you might have some confusion here. Thank you for mentioning the paper [1]. This paper [1] is already cited in our paper. We did try the hyperparameters of the paper [1] but the performance of $\epsilon$-greedy based on the hyperparameters of this paper [1] was lower. This is due to the fact that the paper [1] uses a different algorithm (i.e. Rainbow). Note that the Rainbow algorithm contains many different moving parts: multi-step return, distributional reinforcement learning, noisy networks and dueling architecture (this discussion is also currently present in our paper in the Appendix). On the other hand, in our paper we use baseline double Q-learning. The fundamental reason for this is to simply allocate the main focus to one component to be able to decouple the effects of these varying parts of such a complex algorithm to solely observe the effects of exploration.
>
> Thus, the hyperparameters are set to the level of a more recent paper that utilizes baseline double-$Q$ learning [2]. Note that the paper [2] uses $\epsilon$-greedy and the authors of this paper tuned the hyperparameters for $\epsilon$-greedy for double-$Q$ learning in the Arcade Learning Environment 100K benchmark. On the other hand, we did not conduct any hyperparameter tuning for our algorithm MaxMin Novelty. We used the exact same hyperparameters of [2] for both of the algorithms ($\epsilon$-greedy and MaxMin Novelty). Thus, we simply used hyperparameters tuned for $\epsilon$-greedy for both MaxMin Novelty and $\epsilon$-greedy. Given this it is possible to further improve the MaxMin Novelty results by specific hyperparameter tuning for MaxMin Novelty. But we wanted our paper to provide a fair comparison rather than focusing on specific hyperparameter tuning.
>
> [1] When to use parametric models in reinforcement learning? NeurIPS 2019.
>
> [2] Image Augmentation Is All You Need: Regularizing Deep Reinforcement Learning from Pixels, ICLR 2021. [Spotlight Presentation]
>
> 2. *”Guaranteed to converge to an optimal policy”*
>
> If the main concern is theoretical convergence in the tabular setting, it is straightforward to add some noise to MaxMin Novelty to create a "soft" version where with probability $p\cdot\epsilon$ the minimum action is taken and with probability $(1-p)\cdot\epsilon$ a random action is taken where the probability p is close to 1 ($p \approx 1$).
> This noise will ensure that all actions have some chance of being taken, ruling out the particular concerns regarding the bandit settings, while putting larger weight on the minimum value action in order to improve exploration as described in our paper.
> As the results of the extensive experiments provided in the paper demonstrate, we did not find any issues with convergence in deep reinforcement learning.
> We would like to emphasize that one of the main intuitions behind our method is that taking the minimum action will on average result in higher temporal difference loss than the temporal difference loss obtained by taking a random action. The "soft" version of MaxMin Novelty described right above would also benefit from exactly this intuition, while maintaining theoretical convergence guarantees in the tabular setting if desired.
>
>
> We believe our response targets your questions. We would appreciate it if you could confirm for us that this is the case and would be happy to answer any further questions that you might have.

---

> ### Author Response · Authors · 2022-11-18
> **A new Section in the Appendix**
>
> We have added a new section in the appendix dedicated to extending the MaxMin Novelty algorithm to different deep reinforcement learning algorithms. The human normalized scores across all the tasks of the Arcade Learning Environment 100K benchmark for a baseline distributional reinforcement learning algorithm reported in Appendix Section A4 are substantially higher than data-efficient Rainbow$^*$ [1]. These results once more demonstrate that MaxMin Novelty is able to improve the performance of baseline algorithms beyond that of much more complicated algorithms.
>
> [1] When to use parametric models in reinforcement learning? NeurIPS 2019.
>
> $^*$ Note that Rainbow composes several different advanced techniques: multi-step return, noisy networks and dueling architecture on top of distributional reinforcement learning.

---

> ### Comment · Area_Chair_6QgP · 2022-11-24
> **Thank you! Are you satisfied by the answers?**
>
> Dear reviewer,
>
> Thanks again for your detailed review! The authors have replied back to you. Please read them carefully, and acknowledge their response. If there is still an unclear point about the paper or you do not agree with some of the responses, please let them know. We would like to have a robust discussion now.
>
> If you have any further questions from them, please ask them now. We have to make the final decision soon.
> Also as a courtesy to the authors, please acknowledge their rebuttal.
>
> Thank you,
> Area Chair

---

### Official Review · Reviewer_S1TB · 2022-10-27

**Confidence:** 3
**Correctness:** 2
**Technical Novelty And Significance:** 3
**Empirical Novelty And Significance:** 2
**Recommendation:** 5

**Clarity, Quality, Novelty And Reproducibility:**



The main idea of the paper is hard to follow. And there is some confusing part in the theorems, experiments, and algorithms.

The code of this paper is not open-sourced. Supplying code could help the readers understand the details of the paper.



**Details Of Ethics Concerns:**



**Strength And Weaknesses:**



Strength:

It's interesting to see that using the minimum-valued action could lead to better exploration. The proposed method is very simple and requires minor modifications to existing methods.



Weakness:

Generally, it's hard for me to follow the motivation for using minimum-valued action. The paper seems to claim the minium-valued action should be the one that maximizes the novelty. I didn't see the definition of novelty in Section 3. Although the paper provides several theorems in Sec 3, it's hard to get the main idea of these theorems.

The improvements in Figure 2 are not very significant. Moreover, on some tasks, the paper demonstrates results with 200 million frame training, while on other tasks, the paper demonstrates result with only 50 million frame training (e.g. Enduro). This is very misleading. Did the proposed method converge on these incomplete-reported tasks?





Several Questions:

Algorithm 1 seems quite unclear to me.

What do you mean by Populating the Experience Replay Buffer?

Do you mean you need the MDP model (including the transition and reward function)?

How to obtain $s_{t+1}^{min}$ based on $a^{min}$ (and how to obtain $s_{t+1}$)?

What do you mean by $e$?

**Summary Of The Paper:**

This method proposed to use minimal action with the lowest Q value to maximize novelty. The experiment demonstrates the effectiveness of the proposed methods.



**Summary Of The Review:**



This paper tried to propose a new novelty maximization method with minor computation costs. However, the main motivation for using minimal-valued action is hard to follow. There are some confusing parts in the theorems, experiments, and algorithms that make it hard to assess the idea.

---

> ### Author Response · Authors · 2022-11-15
> **Author Response**
>
> Thank you for the time you have allocated in reviewing our paper.
>
> 1. *’What do you mean by Populating the Experience Replay Buffer?”*
>
> In off-policy methods (e.g. deep $Q$-networks) the experience replay mechanism is used (please see the relevant work [1,2,3,4,5]). Thus, populating the experience replay buffer means the collection of the experiences.
>
> [1] Human-level control through deep reinforcement learning, Nature 2015.
>
> [2] Deep Reinforcement Learning with Double Q-learning, AAAI 2016.
>
> [3] Dueling Network Architectures for Deep Reinforcement Learning, ICML 2016.
>
> [4] Bootstrapped Meta-Learning. ICLR 2022.
>
> [5] Distributional Reinforcement Learning with Quantile Regression, AAAI 2018.
>
>
>
> 2. *”Do you mean you need the MDP model (including the transition and reward function)?”*
>
> Can you elaborate what you mean here as we need the MDP model?
>
>
>
> 3. *”How to obtain $s_{t+1}^{min}$ based on $a_{min}$ and how to obtain $s_{t+1}$?”*
>
> $s_{t+1}^{min}$ simply refers to the state observed after taking action $a_{min}$ in state $s_t$, and $s_{t+1}$ refers to the state observed after taking action $a_t$ in state $s_t$. This information is also currently present in Proposition 3.1.
>
>
>
> 4. *“What do you mean by e?”*
>
> e refers to an episode.
>
>
>
> 5. *“Novelty”*
>
> Novelty here refers to obtaining experiences that yield higher information gain, i.e. the experiences that yield higher temporal difference loss. This is implicitly present in the paper, but we can make this more explicit.
>
>
>
> 6. *”Section 3”*
>
> Section 3 is dedicated to providing theoretical motivations and justifications for why such a method would and should work. In particular, Proposition 3.1 and Proposition 3.2 demonstrate that taking the action that minimizes the state action value function in a given state will yield **higher temporal difference loss** than the temporal difference loss of taking a random action in relationship to disadvantage gap, un-informativeness and smoothness of the state-action value function. While these propositions explain the conditions in which the temporal difference loss obtained by taking the action that minimizes the state-action value function is higher than the temporal difference loss obtained by taking a random action, Section 5 and Section 6 demonstrate that indeed, in deep reinforcement learning in the Arcade Learning Environment, MaxMin Novelty achieves significantly higher performance (i.e. human normalized median scores over all games in the Arcade Learning Environment 100K benchmark).
>
>
>
>
> 7. *”The paper demonstrates result with only 50 million frame training for Enduro”*
>
> This is not an incomplete-reported task. It is a standard way of zooming to where the learning curves experience a jump, and furthermore, this is commonly used in exploration papers to show the sample efficiency gained by the exploration technique. Nonetheless, to obtain a better and stable view of the algorithm performance please look at the human normalized scores reported in the paper in Table 3.

---

> ### Comment · Area_Chair_6QgP · 2022-11-24
> **Thank you! Are you satisfied by the answers?**
>
> Dear reviewer,
>
> Thanks again for your detailed review! The authors have replied back to you. Please read them carefully, and acknowledge their response. If there is still an unclear point about the paper or you do not agree with some of the responses, please let them know. We would like to have a robust discussion now.
>
> If you have any further questions from them, please ask them now. We have to make the final decision soon.
> Also as a courtesy to the authors, please acknowledge their rebuttal.
>
> Thank you,
> Area Chair

---

### Official Review · Reviewer_P9oC · 2022-10-31

**Confidence:** 4
**Correctness:** 3
**Technical Novelty And Significance:** 3
**Empirical Novelty And Significance:** 1
**Recommendation:** 5

**Clarity, Quality, Novelty And Reproducibility:**

The paper is there or thereabouts in terms of clarity and quality, though I’d like to see resolution on at-least some of the aforementioned issues. Leveraging state-transition novelty or “surprise” for exploration is a well-researched area, but the paper’s approach of using the minimum-value action and the supporting propositions are novel, to the best of my knowledge. In terms of reproducibility, the authors have included the hyperparameters and the architecture details, but I didn't find a link to the anonymized code.


**Strength And Weaknesses:**

**Strengths:**

The paper makes an interesting connection between the expected TD errors under the minimum-value action and the uniform action distribution (proposition 3.1). The proposed exploration strategy is computationally lightweight and, on the evidence of the evaluation section, seems to improve the sample efficiency in several benchmarks, especially in the limited interaction setting.

**Concerns to address:**

I have concerns about a couple of details mentioned in the paper and the overall presentation. I request the authors to address the following:
1.  As a conclusion to Proposition 3.1, it is mentioned that the expected TD-error achieved by minimum-value action is higher than that achieved by uniform action distribution. Does this require the entity $D(s) – 2\delta - \eta$ to be non-negative, and if yes, does that somehow follow from the definitions?
2.  Propositions 3.1 and 3.2 are fairly similar in terms of the idea and the proof. I’d recommending moving the Double-Q equations to the appendix to improve the reading experience.
3.  Algorithm 1 box – could you clearly demarcate (with colors preferably) the difference between standard DQN with $\epsilon$-greedy and MaxMin-Novelty? Are there differences both in the experience-generation and the gradient-calculation aspects? Also, there seems to be some notation errors in the TD definition in the box.
4.  Motivating example (Section 4) – when there are just 2 actions, does the difference between $\epsilon$-greedy and MaxMin-Novelty reduce to just the value of $\epsilon$? Could you add $\epsilon$-greedy with different values of $\epsilon$ to Figure 1?
5.  In Section 6, it’s not clear if the TD values are computed for all transitions, or only for the transitions with the exploratory action (taken with probability $\epsilon$).
6.  It would be good to comment on the limitations of the current method, and/or future work. Any thoughts on extensions to continuous-action environments?

**Summary Of The Paper:**

The paper proposes a method for efficient exploration in discrete-action MDP environments. The intuition is to incentivize the agent to take actions that maximize “novelty”, which is measured in terms of the temporal difference (TD) error from the transition. The authors show that under certain assumptions, the expected TD error achieved by the minimum-value action (the actions that minimizes the Q value) is higher than the expected TD error achieved under a uniform action distribution. Based on this motivation, a simple alteration to the $\epsilon$-greedy exploration is proposed where, instead of sampling a uniform action with probability $\epsilon$, the minimum-value action is taken with probability $\epsilon$. Experiments on ALE and ALE 100K benchmarks show that the proposed algorithm compares favorably to the baselines ($\epsilon$-greedy, NoisyNetworks) in terms of sample efficiency due to improved exploration.

**Summary Of The Review:**

My initial rating is “5: marginally below the acceptance threshold”. The paper introduces a simple and interesting idea to facilitate exploration and improve the sample efficiency in deep-RL. The results are encouraging. However, I have some concerns about the material presentation and would be happy to reconsider after discussion with the authors.

---

> ### Author Response · Authors · 2022-11-09
> **Author Response**
>
> Thank you very much for the time you have committed to provide feedback in our paper. Below we have tried to address your questions.
>
> 1. *”The entity $D(s)-2\delta-\eta$ is non-negative?”*
>
> The intuition that the action minimizing the state-action value function for a given state achieves a higher temporal difference loss does require that $D(s) - 2\delta -\eta$ is positive as Proposition 3.1. suggests. The results reported in Figure 4 and Figure 5 demonstrate that $D(s)-2\delta-\eta$ is indeed positive, and thus the temporal difference loss obtained by MaxMin Novelty is higher.
>
> 2. *”Propositions 3.1 and 3.2 “*
>
> If this would help the readability and clarity we would be happy to move this part to the appendix. The reason we kept Proposition 3.2. in the main paper is because double-$Q$ learning is used in the experimental section.
>
> 3. *”Algorithm 1”*
>
> We can include coloring here. Currently, in Algorithm 1 line 8 and line 9 in Column 1, and line 6 and line 7 in column 2 highlight the difference between $\epsilon$-greedy and MaxMin Novelty. Regarding your question, the experience generation aspect is the only difference with standard Q-learning. The description of the algorithm is meant to demonstrate how this difference will subsequently affect the temporal difference loss, and hence the gradients.
>
> 4. *“Could you add $\epsilon$-greedy with different values of $\epsilon$ to Figure 1?”*
>
> Now, we have added a new section to the appendix for different values of epsilon with a variant of the chain-MDP that has more than two actions. The results are reported with varying values of $\epsilon$, and the results in this section once more demonstrate that MaxMin novelty has significantly better performance than the baseline exploration methods.
>
> 5. *’TD values”*
>
> TD values are computed for all transitions.
>
> 6. *”Future extensions”*
>
> Thank you for asking this. Yes, we also have been marinating on some ideas on the extensions of our method to the continuous action domain. We also think there are still interesting open questions in terms of exploration in on-policy algorithms considering the exploration efforts in the off-policy algorithm family. We definitely think this is an interesting research direction, and we hope that our results can contribute to this future research direction.

---

> ### Author Response · Authors · 2022-11-18
> **A Gentle Reminder**
>
> We would like to thank you once more for the time you have invested providing feedback for our paper. The author response period will be ending today. If it is no trouble, would there be a possibility for you to let us know if your questions have been addressed?

---

> ### Comment · Area_Chair_6QgP · 2022-11-24
> **Thank you! Are you satisfied by the answers?**
>
> Dear reviewer,
>
> Thanks again for your detailed review! The authors have replied back to you. Please read them carefully, and acknowledge their response. If there is still an unclear point about the paper or you do not agree with some of the responses, please let them know. We would like to have a robust discussion now.
>
> If you have any further questions from them, please ask them now. We have to make the final decision soon.
> Also as a courtesy to the authors, please acknowledge their rebuttal.
>
> Thank you,
> Area Chair

---

### Author Response · Authors · 2022-11-18
**Summary of the Author Response and the Updated Paper**

We would like to thank all the reviewers for providing feedback on our paper. Below we summarize the highlights of the author response and the changes made to our paper.

- We have added a new Section in the Appendix that is dedicated to providing more results with varying $\epsilon$ regarding the motivating example. **[Appendix Section A3]**
- We have added a new Section to the Appendix demonstrating the efficiency of MaxMin Novelty in different deep reinforcement learning algorithms. In particular, the human normalized median scores of a baseline distributional reinforcement learning algorithm with MaxMin Novelty are **substantially higher** than data-efficient Rainbow, demonstrating that MaxMin Novelty enables baseline algorithms to gain performance to a level that is substantially higher than much more complicated algorithms (i.e. data-efficient Rainbow). **[Appendix Section A4]**
- We have discussed the conditions for which MaxMin Novelty will be guaranteed to converge in the bandit settings, and furthermore we highlighted that as the extensive experiments demonstrate MaxMin Novelty obtains higher performance than prior methods across different reinforcement learning algorithms and across many tasks in the Arcade Learning Environment.

---

### Decision · Program_Chairs · 2023-01-20

**Decision:**

Reject

**Justification For Why Not Higher Score:**

Unclear writing and motivation. The algorithm is not sound. The proof has an error.

**Justification For Why Not Lower Score:**

N/A

**Metareview: Summary, Strengths And Weaknesses:**

The paper proposes a simple exploration strategy in which the action with the minimum action-value function is occasionally selected. Some theoretical results are provided as well as empirical ones.

The reviewers have raised several concerns about this paper, including the followings:
- The clarity and presentation of the paper need improvement. Some reviewers had problem understanding the motivation for using minimum-valued action.
- Correctness of the method is questionable. Reviewers 7Z6y and 3c9Y constructed simple finite armed bandit examples that show the method does perform correctly as an exploration strategy.
- There were concerns regarding the experimental results.


Regarding the correctness, the authors argued that as Deep RL is more complicated than bandits, a strategy that works for deep RL does not need to work with bandits.

I do not agree with this direction of argument. I agree that it is likely that a strategy that works for bandit may not work for deep RL, but a method that is supposed to be able to solve a harder problem, yet cannot solve a simpler problem, is questionable.

Even though we had some discussions between one of the reviewers and the authors, the rest of the reviewers unfortunately did not participate in the discussions. Therefore, I read the paper myself before closely looking at the reviews.

I agree with the issue of clarity, especially in Section 3. The algorithm's motivation is a bit hand wavy. The theoretical results, even though they seem to show some specific and concrete statements, do not really clarify why choosing the action with the minimum action-value is good for exploration-exploitation tradeoff.
The justification that it leads to a larger TD loss, and that maximizes the novelty, which is supposedly good for exploration, is questionable to me. One reason is that what the authors call the TD loss is actually TD error, so it can be both positive and negative. There is no absolute value or squared error of TD error. What is the significance of a very large negative TD error, which makes it a good criteria? Why not choosing an action with a very large positive TD error? It is more intuitive to consider the absolute (or squared) value of TD error as an indicator of surprise and novelty. Even in that case, why is it a good criterion to optimize for exploration-exploitation? This link is not made clearly in the paper.

Perhaps more importantly, I believe the proof of Proposition 3.1 has an error. To see why, let us first consider Definition 3.2. That definition says that for a fixed $\hat{a}$, how different the $\max_a Q_\theta(s,a)$ is from $\max_a Q_\theta(s',a)$, when $s'$ is sampled from the transition kernel $T(s,\hat{a}, \cdot)$. The important point here to notice is that $\hat{a}$ is fixed and does not depend on theta.

Now consider the first inequality in the proof of Proposition 3.1. Let us pay attention to how random variables in the expectation of the first line and the second line are sampled.
First, theta is sampled. This defines $Q_\theta$. As a result, $a^\min$ is selected. As $a^\min$ depends on $Q_\theta$, it is a function of theta, i.e., we have $a^\min(\theta)$. Afterwards, $s^\min_{t+1}$ is selected according to $T(s_t, a^\min(\theta), \cdot)$. This depends on theta because of the dependence of $a^\min$ on $\theta$. So, for each sample of theta, we have a different action $a^\min$ that defines the transition.

Comparing this with Definition 3.2 shows the error in the proof. The definition is based on a fixed $\hat{a}$ across all samples of $\theta$, but we have a theta-dependent action in this proof.
That definition, as is stated, cannot be used to upper bound the difference between two quantities on the first and second line, with an addition of a delta term.

How can this be fixed?
Perhaps the definition should be changed to hold for any $\theta$ in the $\Theta$ space, as opposed to in expectation. I let the authors figure this out and discuss whether that is a reasonable assumption or not.

Overall, because of all these issues, unfortunately I do not think the paper in its current form should be accepted.